# Light-Dependence of Formate (C_1_) and Acetate (C_2_) Transport and Oxidation in Poplar Trees

**DOI:** 10.3390/plants11162080

**Published:** 2022-08-09

**Authors:** Kolby J. Jardine, Joseph Lei, Suman Som, Daisy Souza, Chaevien S. Clendinen, Hardeep Mehta, Pubudu Handakumbura, Markus Bill, Robert P. Young

**Affiliations:** 1Lawrence Berkeley National Laboratory, Climate and Ecosystem Science Division, Berkeley, CA 94720, USA; 2Forest Management Laboratory, National Institute for Amazon Research, Manaus 69067-375, Brazil; 3Environmental Molecular Sciences Laboratory, Pacific Northwest National Laboratory, Richland, WA 99354, USA

**Keywords:** fermentation, C_1_ pathway, alternate respiratory substrates, trees, xylem injection, day respiration, glyoxylate cycle

## Abstract

Although apparent light inhibition of leaf day respiration is a widespread reported phenomenon, the mechanisms involved, including utilization of alternate respiratory pathways and substrates and light inhibition of TCA cycle enzymes are under active investigation. Recently, acetate fermentation was highlighted as a key drought survival strategy mediated through protein acetylation and jasmonate signaling. Here, we evaluate the light-dependence of acetate transport and assimilation in *Populus trichocarpa* trees using the dynamic xylem solution injection (DXSI) method developed here for continuous studies of C1 and C2 organic acid transport and light-dependent metabolism. Over 7 days, 1.0 L of [^13^C]formate and [^13^C_2_]acetate solutions were delivered to the stem base of 2-year old potted poplar trees, while continuous diurnal observations were made in the canopy of CO_2_, H_2_O, and isoprene gas exchange together with δ^13^CO_2_. Stem base injection of 10 mM [^13^C_2_]acetate induced an overall pattern of canopy branch headspace ^13^CO_2_ enrichment (δ^13^CO_2_ +27‰) with a diurnal structure in δ^13^CO_2_ reaching a mid-day minimum followed by a maximum shortly after darkening where δ^13^CO_2_ values rapidly increased up to +12‰. In contrast, 50 mM injections of [^13^C]formate were required to reach similar δ^13^CO_2_ enrichment levels in the canopy with δ^13^CO_2_ following diurnal patterns of transpiration. Illuminated leaves of detached poplar branches pretreated with 10 mM [^13^C_2_]acetate showed lower δ^13^CO_2_ (+20‰) compared to leaves treated with 10 mM [^13^C]formate (+320‰), the opposite pattern observed at the whole plant scale. Following dark/light cycles at the leaf-scale, rapid, strong, and reversible enhancements in headspace δ^13^CO_2_ by up to +60‰ were observed in [^13^C_2_]acetate-treated leaves which showed enhanced dihydrojasmonic acid and TCA cycle intermediate concentrations. The results are consistent with acetate in the transpiration stream as an effective activator of the jasmonate signaling pathway and respiratory substrate. The shorter lifetime of formate relative to acetate in the transpiration stream suggests rapid formate oxidation to CO_2_ during transport to the canopy. In contrast, acetate is efficiently transported to the canopy where an increased allocation towards mitochondrial dark respiration occurs at night. The results highlight the potential for an effective integration of acetate into glyoxylate and TCA cycles and the light-inhibition of citrate synthase as a potential regulatory mechanism controlling the diurnal allocation of acetate between anabolic and catabolic processes.

## 1. Introduction

Aerobic respiration is a biochemical process requiring oxygen that occurs in living plant cells and provides usable chemical energy (adenosine triphosphate), reducing power (nicotinamide adenine dinucleotide), and carbon skeletons (TCA cycle metabolites) needed in numerous physiological processes including growth and development, reproduction, tissue maintenance/defense during stress, and senescence [1]. In higher plants, mitochondrial aerobic respiratory activity is also known to be critical for optimizing photosynthetic metabolism, especially during periods of abiotic stress which can lead to over-reduction of mitochondria and/or chloroplasts and excessive production of reactive oxygen species [2]. Fueled by the central C2 metabolite acetyl-CoA, the tricarboxylic acid cycle (TCA) in mitochondria produces respiratory CO_2_ that can be re-assimilated by photosynthesis, or emitted from leaves via stomata to the atmosphere. While acetyl-CoA production from pyruvate decarboxylation catalyzed by mitochondrial pyruvate dehydrogenase (PDH) is the typical pathway for aerobic respiration of non-structural carbohydrates via glycolysis [3], alternative respiratory pathways active during abiotic stress are also under investigation [4]. Two alternative respiratory pathways receiving increasing attention are the oxidative C1 pathway and acetate fermentation (C2). The C1 pathway, whose quantitative importance in plant carbon cycling is largely unknown, starts with the release of methanol from primary cell-walls [5] and ends with the generation of CO_2_ in both mitochondria and chloroplasts during formate oxidation [6]. Recently, plant tissue concentrations of acetate were shown to dramatically increase due to the activation of acetate fermentation during drought stress [7,8]. Acetate accumulation stimulates the jasmonate (JA) signaling pathway mediated by histone H4 acetylation to confer drought tolerance [7]. JA signaling integrates with other phytohormone signaling pathways including salicylic acid, helps to coordinate plant defense responses to a wide array of abiotic and biotic stresses such as herbivory, pathogen infection, wounding, and high temperatures, freezing, drought, and salt stress [9]. Upregulation of acetate fermentation is considered an evolutionarily conserved drought survival strategy, with the amount of acetate produced directly correlating to survival [7]. Water stress was confirmed to enhance the gene expression of specific enzymes involved in acetate fermentation (e.g., pyruvate decarboxylase) and acetate activation to acetyl-CoA (acetyl-CoA synthetase) [10]. While the signaling roles of acetate have been discussed in terms of protein acetylation, its active form as acetyl-CoA is the major respiratory substrate entering the TCA cycle [11] and used in the synthesis of a numerous primary and secondary metabolites including fatty acids, terpenoids, hydrocarbons, polyketides and flavonoids [12]. In contrast to acetate/acetic acid which may travel long distances in plants via the transpiration stream, acetyl-CoA is impermeable to membranes and is generally required to be synthesized within each organelle [12]. Thus, activation of acetate to acetyl-CoA may be a key mechanism of supplying alternative sources of acetyl-CoA to organelles with high catabolic (e.g., mitochondria) and anabolic (e.g., peroxisome, chloroplast) demands. Therefore, acetyl-CoA, the biologically active form of acetate, links anabolic and catabolic processes and coordinates metabolism with cellular signaling by mediating protein acetylation and serving as a key substrate for biosynthetic and respiratory reactions [12].

Leaf respiration in C3 plants during the day has been widely discussed as light-inhibited relative to the dark [13,14], although this change in respiratory flux is uncertain because of difficulties in separating simultaneous CO_2_ sources (e.g., respiration and photorespiration) and CO_2_ sinks (e.g., photosynthesis) [13]. The biochemical causes of the apparent light suppression of leaf respiration are under active investigation. Mechanisms, include a light inhibition of key enzymes including mitochondrial pyruvate dehydrogenase (PDH) and citrate synthase, a decrease in the photochemical efficiency of photosystem II with irradiance, an increasing proportion of photorespiration with irradiance as well as refixation of internal CO_2_ sources (respiration, photorespiration, C1 pathway, etc.) by photosynthesis [15,16]. In addition, coordinated interactions of respiratory substrates with other biochemical pathways during the day may also be involved [15,16]. For example, biosynthetic pathways active during the day such as light-dependent de novo fatty acid biosynthesis in chloroplasts, which utilize identical substrates as mitochondrial respiration (e.g., acetate/acetyl-CoA) [17]. Leaf day respiration involves many biochemical and enzymatic reactions that interact with light [17,18]. Through its impact on enzyme activity, light is known to exert a strong control over respiratory substrate allocation to biosynthetic versus catabolic metabolism [19]. For example, plant cells have both a mitochondrial and plastidal form of pyruvate dehydrogenase (PDH) that catalyzes the oxidative decarboxylation of pyruvate to acetyl-CoA. The mitochondrial form is highly active in the dark compared with the light and generates acetyl-CoA for use in the TCA cycle during aerobic respiration. In contrast, the plastidal form of PDH provides acetyl-CoA for de novo fatty acid biosynthesis with a higher activity in the light compared with the dark [3]. Consistent with the view of the differential light controls over the mitochondrial and chloroplast forms of PDH, a recent study concluded that light inhibition of the mitochondrial form of PDH can largely explain the apparent light suppression of leaf day respiration relative to dark respiration in sunflower [20].

In addition to the PDH reaction which generates acetyl-CoA from pyruvate, light inhibition of citrate synthase (CS), which condenses acetyl-CoA with oxaloacetate as the first step of the TCA cycle in mitochondria, has also been demonstrated [21]. However, the quantitative importance of CS light inhibition in suppressing TCA cycle decarboxylations in the light remains unclear. In order to bypass the PDH reaction and evaluate potential light inhibition of the CS reaction, in this study, we supplied acetate via the transpiration stream to detached and intact poplar trees. We hypothesized that acetate in the transpiration stream of trees can be used as a respiratory substrate in leaves via activation to acetyl-CoA and that acetate allocation to the TCA cycle in the light (day respiration) is suppressed by CS light inhibition. To evaluate the possibility of CS light inhibition, we bypassed the normal production of acetyl-CoA via the mitochondrial PDH reaction by supplying external [^13^C_2_]acetate via the transpiration stream and quantified diurnal respiratory emission of ^13^CO_2_.

Photosynthesis with atmospheric ^13^CO_2_ is regularly used to trace plant and ecosystem fates of recently assimilated carbon, including as substrates for stem and soil respiration [22]. However, novel approaches are being investigated including the direct injection of isotopically labeled substances into the xylem of trees including D_2_O [23], H_2_^18^O [24], and H^13^CO_3_ [23]. For example, xylem injections of D_2_O have been used to characterize whole-tree water transport and storage properties [21]. Direct xylem injections of H^13^CO_3_ have been described as an alternative method to atmospheric ^13^CO_2_ used to determine the production and fate of recently fixed photosynthate in trees [25]. Stem injections of ^13^C-organic acids such as aspartic acid have been used to probe above–below ground carbon exchange [26]. However, there is a general lack of studies in trees reporting stem injections of respiratory substrates such as sugars and alternative respiratory substrates such as C1 (formate) and C2 (acetate) carboxylic acids with parallel studies on transport and canopy leaf respiratory processes. Moreover, injection studies typically perform rapid ‘pulse’ injections at one or several points in time with high concentrations of ^13^C-labeled substrates delivered over a short period of time. Continuous flow-controlled injections of ^13^C-labeled central plant metabolites would allow for quantitative in situ studies of their fates in plants including transport and integration into specific biochemical pathways and their biological and environmental dependencies.

In this study, we first hypothesize that two alternative respiratory pathways, including the oxidative C1 pathway and acetate fermentation (C2) are active in leaves and can be supported by formate (C1) and acetate (C2) derived from other tissues (e.g., roots and stems) and transported to leaves via the transpiration stream. Secondly, we hypothesize that due to the light inhibition of key respiratory enzymes such as CS, light impacts the allocation of leaf acetate (C2) between anabolic (e.g., lipid biosynthesis) and catabolic (e.g., mitochondrial respiration) metabolism. We test these hypotheses in potted *Populus trichocarpa* trees by developing and applying the dynamic xylem solution injection (DXSI) method for continuous diurnal injections of [^13^C]formate and [^13^C_2_]acetate solutions into the xylem of poplar trees with controlled liquid injection flow rates together with continuous branch gas exchange measurements in the canopy of transpiration, net photosynthesis, isoprene emissions, and δ^13^CO_2_. These whole tree results are compared with leaf gas exchange observations on detached poplar branches pretreated with 10 mM [^13^C]formate and [^13^C_2_]acetate solutions and subjected to repeated light–dark cycles under controlled environmental conditions. The results are discussed in terms of the different timescales of formate and acetate metabolism during transport in the transpiration stream of poplar trees and the apparent impact of light on regulating acetate allocation in leaves towards respiratory versus anabolic metabolism. We conclude by discussing the potential broader applications of the DXSI method in plant research.

## 2. Results

### 2.1. Leaf Concentrations of TCA Cycle Intermediates and Jasmonates

Acetate delivery to detached poplar branches (*Populus trichocarpa*) via the transpiration stream enhanced leaf concentrations of TCA cycle intermediates and jasmonates (Figure 1). Relative to water-fed controls, acetate-fed poplar branches showed large statistically significant increases in TCA cycle intermediates citric acid (301 +/− 192%) and succinic acid (180 +/− 56%). However, the increase in the mean concentrations of α-ketoglutaric acid was not statistically significant due to the relatively high variation observed between the three leaf samples (725 +/− 652%). Moreover, leaf concentrations of the phytohormone dihydrojasmonic acid also showed strong increases in acetate-treated leaves (882 +/− 455%). This is consistent with the role of acetate in integrating into the TCA cycle and activating the jasmonate signaling pathway via protein acetylation [7].

### 2.2. Dynamic Xylem Solution Injection (DXSI) System for Continuous Whole Tree Injections of [^13^C_2_]acetate and [^13^C]formate

The DXSI system was installed in the stem of individual potted poplar trees transferred from the greenhouse to the laboratory under automated soil watering and controlled lighting conditions (Figure 2). The DXSI experiments were used to quantify diurnal transport and oxidation patterns of C1 and C2 organic acids as alternative respiratory substrates in the transpiration stream of poplar trees together with leaf respiratory metabolism via dynamic branch gas exchange observations of transpiration, net photosynthesis, isoprene emissions, and δ^13^CO_2_. The higher solution injection flow rate into the xylem during the day (150 µL min^−1^) compared with the night (70 µL min^−1^) was based on the observations of a strong circadian pattern in branch stomatal conductance and transpiration patterns. Non-zero elevated branch transpiration values occurred at night associated with the lack of complete stomatal closure in the dark. However, transpiration increased during the day together with net photosynthesis and isoprene emissions reaching maximum values around mid-day (Figure 3a). Stem base injection of 10 mM [^13^C_2_]acetate started in the day around 11:00 a.m. and induced an overall pattern of canopy branch headspace ^13^CO_2_ enrichment over the next 7 days. Starting with a δ^13^CO_2_ value of the reference air entering the branch enclosure (−7.0‰ to −10.0‰) a strong ^13^C-enrichment in headspace CO_2_ could be detected on the first day following the initiation of the xylem injections. However, despite the decrease in transpiration rates during night 1, δ^13^CO_2_ values continued to increase in the dark, reaching a maximum around 0‰ around midnight. Upon the light switching on the following day, branch transpiration increased together with a marked suppression in daytime δ^13^CO_2_ which rapidly increased by +7.0‰ (from +1.0‰ to +8.0‰) upon darkening. While there was an overall pattern of ^13^C-enrichment in CO_2_ over the next 6-days (δ^13^CO_2_ up to +27‰), the same pattern emerged with a strong diurnal structure in δ^13^CO_2_ reaching a mid-day minimum followed by a maximum shortly after darkening where δ^13^CO_2_ values rapidly increased by up to 12‰. Therefore, branch headspace δ^13^CO_2_ was enhanced at night and suppressed during the day, despite the higher transpiration rates during the day.

In contrast to DXSI ^13^C-labeling with a 10.0 mM solution of [^13^C_2_]acetate, injection of a 10 mM [^13^C]formate solution into the xylem of poplar trees resulted in only small ^13^C-labeling of headspace CO_2_ of canopy branches (Appendix A). Canopy branch transpiration, net photosynthesis, and isoprene emissions showed strong diurnal patterns with a mid-day maximum. Branch headspace δ^13^CO_2_ values of the reference air entering the branch enclosure (−9.0‰ to −10.0‰) showed only small ^13^C-enrichment on the first day (−3 to −5‰) following the initiation of the [^13^C]-formate xylem injections. Maximum δ^13^CO_2_ values reached around −5.5‰ on mid-day of the first day, together with transpiration, net photosynthesis, and isoprene emissions. In contrast to the [^13^C_2_]acetate injections, injection of a 10 mM [^13^C]formate solution did not result in strong and rapid enhancements in δ^13^CO_2_ upon darkening. In general, δ^13^CO_2_ declined slightly with transpiration during the night, and increased slightly during the day. The highest value of δ^13^CO_2_ was approximately −5.0‰ occurring around mid-day of day 3.

In order to increase the ^13^C-labeling of CO_2_ during the DXSI experiments with [^13^C]formate xylem injections to a more comparable level with those observed under 10 mM [^13^C_2_]acetate injections, a DXSI experiment was performed starting with 10 mM [^13^C]formate injections for one week followed by a switch to 50 mM [^13^C]formate injections for an additional four days (Appendix A). Similar to the first tree injected with 10 mM [^13^C]formate injections for one week, headspace CO_2_ showed only small ^13^C-enrichment, with δ^13^CO_2_ remaining below 0.0‰ and showing maximum values during the day when transpiration was higher. Upon switching from the 10 mM [^13^C]formate solution to the 50 mM [^13^C]formate solution around midday on the 8th day, the magnitudes of the diurnal increase in δ^13^CO_2_ were enhanced, reaching maximum values of +20.0‰ around mid-day. These values are more comparable with those observed in the 10 mM [^13^C_2_]acetate injections. Despite the larger amplitude of day–night differences, δ^13^CO_2_ still followed diurnal patterns of transpiration, and did not show strong enhancements following darkening as observed in 10 mM [^13^C_2_]acetate DXSI experiments. When only a 50 mM [^13^C]formate solution was used in a separate DXSI experiment, similar results were obtained with δ^13^CO_2_ generally following diurnal patterns of transpiration without a large increase following darkening at the beginning of the night period (Figure 3b).

### 2.3. Light-Dependence of Headspace δ^13^CO_2_ of [^13^C_2_]acetate and [^13^C]formate Treated Detached Poplar Branches

In order to further evaluate the role of light on the oxidation of acetate and formate in the transpiration stream to CO_2_ in leaves, leaf gas exchange in the light under controlled environmental conditions was quantified from detached branches pretreated with 10 mM [^13^C_2_]acetate (Figure 4a) or [^13^C]formate (Figure 4b). Detached branches pretreated with 10.0 mM [^13^C_2_]acetate in the light (Figure 4a) showed similar levels of leaf headspace ^13^CO_2_-enrichment (δ1^13^CO_2_ up to +20.0‰) to that of the whole plant DXSI experiment with 10.0 mM [^13^C_2_]acetate injections where the dynamic branch headspace δ^13^CO_2_ values reached up to 20.0‰ during the day. Moreover, the strong ^13^CO_2_-enrichment in the dynamic leaf headspace atmosphere, as measured in real time by CRDS, was confirmed using grab samples collected manually and analyzed offline using IRMS. Throughout the 4-h experiment in the light, net photosynthesis, stomatal conductance, and transpiration slightly decreased, but maintained values typical of physiologically active mature poplar leaves in the light. In contrast to the whole plant DXSI injections with 10 mM [^13^C]formate, which resulted in only small ^13^C-labeling of headspace CO2 of canopy branches, detached branches pretreated with 10.0 mM [^13^C]formate in the light showed much higher levels of leaf headspace ^13^CO_2_-enrichment (δ^13^CO_2_ up to +322‰) (Figure 4b). Throughout the 4-h experiment in the light, net photosynthesis, stomatal conductance, and transpiration maintained high values representative of highly physiologically active mature poplar leaves in the light. In summary, leaf δ^13^CO_2_ values in 10.0 mM [^13^C_2_]acetate-treated branches in the light reached similar values as branches of intact poplar trees during DXSI experiments with 10.0 mM [^13^C_2_]acetate injected into the xylem. In contrast, leaf δ^13^CO_2_ values in 10.0 mM [^13^C]formate-treated branches in the light far exceeded values from canopy branches during DXSI experiments with 10.0 mM [^13^C]formate injected into the xylem.

To evaluate the role of light on leaf ^13^CO_2_ production from [^13^C_2_]acetate oxidation via mitochondrial respiration, a leaf from a branch pretreated with 10.0 mM [^13^C_2_]acetate was placed in the dynamic leaf chamber in the light and subjected to three sequential light–dark and dark–light transitions under controlled environmental conditions. Leaf headspace δ^13^CO_2_ on branches pretreated with 10.0 mM [^13^C_2_]acetate showed a rapid and strong response to repeated light–dark and dark–light cycles (Figure 5). Consistent with the whole plant and branch gas exchange studies, switching the light off at the leaf level induced a rapid increase in δ^13^CO_2_ by up to +70‰ within 6 min after darkening. Upon switching the light back on, δ^13^CO_2_ rapidly decreased by −45‰ within 6 min after re-illumination, and thereafter decreased more slowly to return to close to the initial δ^13^CO_2_ value after 30 min. Three sequential dark–light and light–dark cycles showed similar results, demonstrating the reversibility of the apparent light-inhibition of [^13^C_2_]acetate respiration (Figure 5).

## 3. Discussion

### 3.1. Dynamic Xylem Solution Injection (DXSI)

Introduction of solutions into plants and trees have long been of interest to farmers, forest managers, land owners, and researchers with injections directly into the conductive tissues of trees first investigated systematically by Leonardo da Vinci [27].

Two main types of solution delivery methods have been developed including infusion where low solution flow rates are delivered over long time periods [28], and direct injection of liquid solutions into the xylem [29]. In this study, by controlling the injection flow rate into the xylem of two-year old poplar trees at low values (e.g., 70–150 µL min^−1^), the DXSI injection method allows for continuous injections of solutions with programmable delivery rates such that high volumes (e.g., L) of solution can be injected over longer periods of time. Thus, the developed DXSI method presented here has features of both the infusion method and the injection methods and is therefore ideal for quantitative studies of leaf diurnal respiration patterns using isotopically labeled substances. While we held DXSI injection flow rates constant during the day and night, a circadian rhythm of transpiration and stomatal conductance was observed, despite constant environmental conditions. Thus, we suggest future improvements of the DXSI method to continuously scale injection flow rates with sap flow rates. This would, in theory, enable a constant concentration of injected substances in the xylem sap, regardless of the sap velocity. Thus, the DXSI method is anticipated to be useful as a continuous injection technique allowing the observation of diurnal patterns not typically resolved by rapid pulse-injection experiments. While we used ^13^C-solutions for isotopic labeling, this technique could also be applied to plant hormones and other biologically active substances where continuous injections of low concentrations would be useful for studying transport, metabolism, and biological impacts. In addition to ^13^C-labeled respiratory substrates described here, another promising avenue is the partitioning of net leaf O_2_ fluxes in the light into gross O_2_ fluxes associated day respiration, photorespiration and photosynthesis in leaves by combining the DXSI technique with H_2_^18^O xylem injections and leaf oxygen flux and δ^18^O_2_ measurements [30].

### 3.2. Transport and Metabolism of C1 and C2 Organic Acids in Detached Branches and Whole Trees

Numerous analytical methods are currently being explored in the study of leaf day respiration and the apparent suppression effects of light [17]. The two main approaches used to estimate leaf day respiration rates are the Loreto ^13^CO_2_ isotopic method [31] and the Kok method [13]. The Loreto method assumes that in an atmosphere of ^13^CO_2_, all ^12^CO_2_ detected is from respiration, and is considered to be the most accurate method [17]. The Kok method uses the abrupt change in the initial slope of the response of net photosynthesis to irradiance at low light (i.e., sudden change in the quantum yield of photosynthesis near the light compensation point) [32,33,34]. Studies using stable isotope tracing have demonstrated important insights when combined with real-time gas exchange and leaf physiological measurements. For example, leaf labeling with ^13^CO_2_ in the atmosphere and [^13^C]glucose and [^13^C]pyruvate leaf feeding have provided strong evidence for the reversible inhibition of respiratory enzymes in illuminated leaves [18,35]. For instance, [^13^C]glucose was previously shown to be blocked in the light from entering glycolysis and instead diverted to sucrose synthesis [34]. More recently leaf [^13^C]glucose decarboxylations have been quantified as low, but significant in the light, and increase with decreasing irradiance below 100 µmol photons m^−1^ s^−1^ [20]. Moreover, while TCA cycle decarboxylations of [^13^C]pyruvate are strongly light-inhibited (e.g., 80–95%), mitochondrial PDH decarboxylating activity has been shown to be only ~30% inhibited in the light relative to the dark [33,34]. Thus, the strong suppression of TCA cycle decarboxylations in the light may be related to the strong light inhibition of other TCA cycle enzymes such as citrate synthase (CS), which condenses acetyl-CoA with oxaloacetate as the first step of the TCA cycle in mitochondria [21].

While light inhibition of CS has been demonstrated [21], its role in the light suppression of leaf day respiration remains under investigation. In this study, we bypassed formation of acetyl-CoA via the mitochondrial PDH reaction, by supplying [^13^C_2_]acetate via the transpiration stream in detached branches and whole trees using a new developed method termed Dynamic Xylem Solution Injection (DXSI). Given that formate oxidation to CO_2_ as a part of the C1 pathway in plants does not involve TCA cycle enzymes [5], we used both detached branch labeling and whole tree DXSI labeling with [^13^C]formate to contrast with [^13^C_2_]acetate transport and light-dependent metabolism in poplar trees. The similar ^13^C-enrichment of headspace CO_2_ in the whole plant DXSI (Figure 3a) and detached branch (Figure 4a) labeling experiments with 10 mM [^13^C_2_]acetate suggests that acetate is efficiently transported in the xylem to leaf cells where it is utilized as a respiratory substrate. However, within minutes of switching off the grow light at the start of each night period during the whole tree DXSI studies (Figure 3a), or the light source within the leaf chamber during the detached branch studies (Figure 5), δ^13^CO_2_ values rapidly increased. Upon switching the light back on in the morning for the whole tree DXSI experiment or after ~30 min of darkness during the detached branch/leaf studies, δ^13^CO_2_ values rapidly decreased, returning to similar values in the previous light period. These observations demonstrate that the light inhibition of leaf ^13^CO_2_ efflux during [^13^C_2_]acetate labeling is reversible.

The results demonstrate that acetate in the transpiration stream is an effective respiratory substrate. However, before acetate can be utilized as a respiratory substrate in mitochondria, it must first be activated to acetyl-CoA. As acetyl-CoA is membrane impermeable, the use of acetate as a respiratory substrate is assumed to require acetate activation to acetyl-CoA within mitochondria. While acetate is known to be activated to acetyl-CoA in peroxisomes and chloroplasts through the reaction catalyzed by an acetyl-CoA synthetase localized to those organelles [36], a mitochondrial acetyl-CoA synthetase has not been clearly demonstrated in plants as it has in animals [37]. Another potential mechanism for acetate utilization in mitochondrial respiration is the glyoxylate cycle, which starts in peroxisomes and depends on mitochondrial TCA cycle metabolism, providing plants the ability to convert C2 acetate to C4 respiratory substrates such as citrate, isocitrate, glyoxylate, malate, and oxaloacetate which can be used to replenish the TCA cycle or as precursors for amino acid or sugar biosynthesis [38]. The glyoxylate cycle in seeds is known for its important roles in converting storage lipids to sugars through the integrated activities of β-oxidation, the glyoxylate and TCA cycles, and gluconeogenesis [39]. In contrast, relatively less is known about the function of the glyoxylate cycle in leaves including its integration with fermentation, photosynthesis, photorespiration, and mitochondrial respiration. However, Arabidopsis thaliana mutant seedlings which lack the glyoxylate cycle are completely unable to convert acetate into sugar and show a dramatic decrease in the frequency of seedling establishment and compromised growth in low light or reduced daylength conditions [40]. In the case where acetate can be activated to acetyl-CoA directly within mitochondria, the observations of a rapid and strong enhancement in leaf ^13^CO_2_ efflux upon darkening during [^13^C_2_]acetate labeling may be related to the dark activation of mitochondrial CS [21]. Similarly, in the case where acetate is activated to acetyl-CoA within peroxisomes as a part of the glyoxylate cycle, peroxisomal CS may also be strongly dark activated. Recent studies with the photosynthetic green alga Chlamydomonas reinhardtii demonstrate that not only do the abundances of enzymes involved in the glyoxylate cycle become elevated under heterotrophic conditions (dark and in the presence of external acetate), peroxisomal CS activity dramatically increases via protein acetylation [41]. Nonetheless, it is important to note that other mechanisms could also be at play, and lead to a similar “apparent” suppression of [^13^C_2_]acetate respiration in the light including the re-assimilation of respiratory ^13^CO_2_ by photosynthesis [12]. However, in contrast to whole [^13^C_2_]acetate labeling, DXSI experiments with 10 and 50 mM [^13^C]formate did not show strong enhancements in δ^13^CO_2_ branch efflux upon darkening at night (Figure 3b and Appendix A). This implies that re-assimilation of respiratory ^13^CO_2_ by photosynthesis may not explain the strong dark enhancement in δ^13^CO_2_ observed during [^13^C_2_]acetate labeling. Nonetheless, the metabolic pathway(s) of [^13^C_2_]acetate oxidation to ^13^CO_2_ including the subcellular location of activation to acetyl-CoA, potential involvement of the glyoxylate cycle in integrating acetate metabolism with the TCA cycle, light inhibition of TCA and glyoxylate cycle enzymes including CS, and the role of re-assimilation of respiratory ^13^CO_2_ by photosynthesis requires further research to resolve.

Supplying 10 mM [^13^C_2_]acetate to detached branches and whole trees during DXSI injections showed similar ^13^C-enrichments in leaf and branch headspace δ^13^CO_2_, respectively. In contrast, detached branch labeling with 10 mM [^13^C]formate resulted in much higher values of leaf headspace δ^13^CO_2_ compared with branch headspace δ^13^CO_2_ during 10 mM [^13^C]formate whole tree DXSI experiments (Figure 4b vs. Appendix A). This suggests that formate is rapidly oxidized to CO_2_ within the transpiration stream, likely catalyzed by formate dehydrogenase enzymes localized to both mitochondria and chloroplasts in higher plants [6]. During whole plant DXSI labeling experiments, a 50 mM [^13^C]formate solution was required to reach similar levels of canopy branch δ^13^CO_2_ as 10 mM [^13^C_2_]acetate (Figure 3b vs. Figure 3a). These observations suggest that formate has a shorter lifetime within the transpiration stream of poplar relative to acetate, such that by the time the solution reaches the canopy leaves, the majority of the [13C]formate has already been oxidized to ^13^CO_2_. Moreover, relative to acetate, formate did not show indications of a strong light-inhibition of oxidation to ^13^CO_2_. This is consistent with the observations that δ^13^CO_2_ during DXSI delivery of 10 and 50 mM [^13^C]formate generally showed maximum values in the light during the day when delivery rates of [^13^C]formate to leaves via the transpiration stream was higher. In contrast, branch headspace δ^13^CO_2_ values during DXSI experiments with [^13^C_2_]acetate generally showed minimum values during the day (Figure 3a). These observations show the utility of combining labeling of detached branches with high environmental control over leaf gas exchange conditions such as light during short-term experiments over hours with whole tree DXSI experiments over longer time scales (e.g., days to months) to characterize the transport and metabolism of respiratory intermediates from small (leaf to branch) to larger (stem and canopy) scales. Thus, reducing the need for destructive sampling, long-term DXSI experiments with 13C-labeled respiratory substrates are anticipated to be a useful tool in the investigation of quantitative day and night leaf respiratory processes including its integration with other major primary metabolic pathways including fermentation, photosynthesis, photorespiration, and the glyoxylate cycle.

## 4. Materials and Methods

### 4.1. Plant Material

We focused on the fast-growing tree genera Populus as it is widely distributed across many temperate and cold regions around the globe and is actively being investigated for afforestation efforts and as renewable sources of biofuels and bioproducts. Moreover, with the completion of genome sequencing, *Populus trichocarpa* is considered highly amendable to genetic engineering studies aimed at altering carbon allocation patterns of woody plants. Poplar (black cottonwood: *P. trichocarpa*) saplings were obtained from Plants of the Wild (Washington State, DC, USA). The two-year old trees were maintained at the UC Berkeley Oxford Tract greenhouse, grown in super soil media (Scotts Co., Marysville, OH, USA) within 5-gallon pots with automated watering. Natural sunlight was supplied and supplemented with additional LED lighting (Lumigrow 325 Pro, Emeryville, CA, USA) controlled using an application controller with photocell (Argus Titan, BC, Canada). The controller was programmed to turn LED lights off when detecting exterior light levels above 850 μmol m^−2^ s^−1^ during the 16-h photoperiod (6 a.m. to 10 p.m.). For each of the four trees studied, they were first transported before sunrise from the UC Berkeley Oxford greenhouse to the nearby analytical laboratory (Lawrence Berkeley National Laboratory) and placed under an LED grow light (90 W Baisheng Semiconductor Lighting Co., Ltd., Shenzhen, China) supplying a constant 400–1000 µmol m^−2 s−1^ photosynthetically active radiation intensity at the top of the tree canopy (depending on position of canopy branch) during the daytime (6:00–18:00).

### 4.2. Dynamic Xylem Solution Injection (DXSI) System

The DXSI system consists of a liquid handling pump interfaced with the solution reservoir (pump input) and the xylem of a poplar tree (pump output) through 1/16′′ outer diameter tubing (Figure 6). For each of the four trees studied, they were first transported before sunrise from the UC Berkeley Oxford greenhouse to the nearby analytical laboratory (Lawrence Berkeley National Laboratory, Berkeley, CA, USA) and placed under an LED grow light (90 W Baisheng Semiconductor Lighting Co., Ltd.) supplying 400–1000 µmol m^−2^ s^−1^ photosynthetically active radiation intensity at the top of the tree canopy during the daytime (6:00–18:00). A single automated watering system was used to deliver water (~200 mL in 10 s) to the soil of the potted tree every 3 h. Prior to the initiation of the light period, two 1/16” silcosteel tubes (1/16” outer diameter and 0.04” inner diameter, Restek Inc., Bellefonte, PA, USA) containing the solution to be injected were installed into the xylem of the tree. A 1/16′′ diameter hole was drilled 2.5 cm deep at a 45-degree angle (with respect to horizontal) approximately 15 cm above the soil surface. Immediately following this, a second hole 18 cm above the soil surface on the opposite side of the first hole was created with the same method. The two 1/16” tubes containing the ^13^C-labeled solution were each inserted 2.5 cm deep directly into the drilled holes and sealed at the surface of the stem using 10 sequential layers of Loctite epoxy adhesive with Loctite adhesive accelerator (Loctite 380 adhesive; Loctite 712 adhesive accelerator, Henkel Corporation, Rocky Hill, CT, USA). Prior to installation, the silcosteel tubes were each gently bent into a half-circle shape, such that the metal tubing could be inserted into opposite sides of the poplar tree stem, allowing for fluid injection into two locations in the xylem. The two silcosteel tubes were connected together with a 1/16” stainless steel tee which was connected to the output of a programmable liquid handling pump (M6 M Series Pump, Valco Instruments Co., Inc., Houston, TX, USA) via a 1/16” Teflon tube. This setup allowed for the liquid flow from the M6 pump to be diverted to both xylem injection points. Prior to installation of the silcosteel tubes into the xylem, the liquid pump and tubing were flushed with the ^13^C-solution overnight (50 µL min^−1^ × 8–10 h).

Over 7 days, 1.0 L solutions of 10.0 mM Na-[^13^C_2_]acetate or 10–50 mM Na-[^13^C]formate were placed in the M6 Pump’s supply reservoir and solution output dispensing volumetric flow rates were programmed using the Mforce controller application (VICI-Valco Instruments). A diurnal dispensing method was used, with a constant xylem injection flow rate of 70 μL min−1 during the night (8:00 p.m. to 6:00 a.m.) and 150 μL min^−1^ during the day (6:00 a.m. to 8:00 p.m.).

### 4.3. Detached Branch Labeling

In order to compare whole tree responses to [^13^C]formate and [^13^C_2_]acetate xylem solution injections, detached branch solution labeling studies were also performed with leaf gas exchange measurements conducted under constant daytime environmental conditions. Poplar trees were obtained from the greenhouse (Oxford Tract Farm, Berkeley, CA, USA) before sunrise and transported to the analytical laboratory. A 30 cm branch was cut from the poplar tree and immediately immersed into a 250 mL flask of solution (10 mM [^13^C_2_]acetate or 10 mM [^13^C]formate) and recut under the solution to prevent excess air from entering and disrupting the transpiration stem of the branch. The detached branch in the solutions was maintained in a growth chamber (Percival Intellus Control System, Perry, IA, USA) with daytime settings of 27.5 °C, 30% light (6:00 a.m.–8:00 p.m.) and 25 °C, 0% light during the night (8:00 p.m.–6:00 a.m.) for one day (10–16 h) prior to leaf gas exchange measurements.

### 4.4. Relative Quantification of Leaf TCA Cycle and Jasmonate in Acetate Treated Leaves

In order to assess the role of acetate on leaf respiration and defense signaling, detached poplar branch feeding with 10 mM acetate solutions were used to characterize acetate-induced changes in leaf TCA cycle intermediates (respiration) and jasmonates (signaling and defense) concentrations using Reverse-Phase Liquid Chromatography Mass Spectrometry (RP LC-MS). Detached poplar branches were placed in water (*n* = 3) or 10 mM acetate (N = 3) for 24 h in a humidified growth chamber (RH: 90–95%) and recut under the solution to prevent excess air from entering and disrupting the transpiration stem of the branch. The detached branch in the solutions was maintained in a growth chamber (Percival Intellus Control System, Perry, IA, USA) with daytime settings of 27.5 °C, 30% light (6:00 a.m.–8:00 p.m.) and 25 °C, 0% light during the night (8:00 p.m.–6:00 a.m.). One leaf from the water control (3) and acetate treated branches (3) were extracted, dried, resuspended in 80:20 methanol:water and analyzed for TCA cycle and jasmonate metabolite levels. Compounds were separated using Thermo Hypersil GOLD (2.1 × 150 mm length, 3 μm particle size) with a 2.1 mm, 0.2 µm filter cartridge, column temperature set to 40 °C and a flow rate of 400 µL min^−1^. Mobile phase A (0.1% Formic acid in H_2_O) and B (0.1% Formic acid in ACN) were initially 90:10, respectively. The gradient continued as follows: 0–2 min 90% A; 2–11 min 10% A; 11–12.5 min held at 10% A; 12.5–13.5 min 90% A with an increased flow rate of 500 µL min^−1^; 13.5–14 min 90% A; 14–14.5 min 90% A with a decreased flow rate of 400 µL min^−1^; 14.5–15 held at 90% A. Data-dependent acquisition was acquired on a Thermo Qexactive Orbitrap mass spectrometer with a scan range of 80 to 800 m/z. Thermo Raw files were processed using MZmine 2 and exported as csv. Citric acid (1.04 min retention time), succinic acid (1.26 min retention time), and α-ketoglutaric acid (1.19 min retention time) were identified and verified using retention time, monoisotopic mass, and MS2 match to internal database. Dihydrojasmonic acid (7.74 min retention time) was identified using only retention time and monoisotopic mass match to internal database.

### 4.5. Branch Gas Exchange Analysis during Whole Plant DXSI Experiments

In order to evaluate in vivo leaf respiratory activity, ^13^C-labeling of leaf respiratory CO_2_ in poplar leaves was achieved by delivery of [^13^C]formate and [^13^C_2_]acetate solutions to whole trees using the Dynamic Xylem Solution Injection (DXSI) system (Figure 2a) and detached branches placed in the respective solutions (Figure 2b). A dynamic 5.0 L Tedlar branch enclosure (5 Liter Tedlar Bag, CEL Scientific Corporation, Los Angeles, CA, USA) was installed on one of the branches near the top of the tree. Gas exchange analysis was performed by enclosing a branch with 5–10 leaves inside the enclosure with 2.0 L min^−1^ hydrocarbon free air continuously passing through. The air contained ambient levels of CO_2_ and H_2_O, with trace volatile organic compounds (VOCs) removed by catalytic oxidation of laboratory air at 400 °C (737 pure air generator, Aadco Instruments Inc, Cleves, OH, USA). The enclosure was secured around the stem at the base with air entering the top through the ¼” valve port and venting at the bottom where the enclosure was secured to the stem (1775 mL min^−1^), with 225 mL min^−1^ diverted to three real-time gas sensors for quantification of (1) CO_2_ and H_2_O branch headspace concentrations (Li7000, Li-Cor Biosciences, Lincoin, NE, USA), (2) ^12^CO_2_ and ^13^CO_2_ branch headspace concentrations together with ẟ^13^CO_2_ (G2131-I High Precision Isotopic Carbon analyzer, Picarro Inc, Santa Clara, CA, USA), and (3) isoprene branch headspace concentrations via high sensitivity quadrupole Proton Transfer Reaction-Mass Spectrometer (PTR-MS, Ionicon, Austria). While emission fluxes of isoprene were calculated from the branches, fluxes of transpiration and photosynthesis were not quantified due to the variability of incoming H_2_O and CO_2_ in lab air.

The Li7000 was operated in absolute mode with the reference cell continuously provided 100 mL min−1 of ultra-high purity nitrogen (5.0 purity, Praxair, Danbury, CT, USA) and designated as 0 ppm CO_2_ and 0 mmol mol^−1^ H_2_O. A 100 mL min^−1^ volume of air from inside the branch enclosure was directed to the Li7000 sample cell with average CO_2_ and H_2_O concentrations recorded every minute. Branch headspace ^12^CO_2_ and ^13^CO_2_ concentrations together with ẟ^13^CO_2_ were analyzed using a G2131-I High Precision Cavity Ringdown Spectrometer (CRDS, Picarro Inc., USA). 25 mL/min of branch enclosure headspace air was passed first through a water trap to remove water vapor (Dririte), before passing through a particle filter (Gelman 1 Micron Filter Assembly) and entering the Isotopic CO_2_ analyzer. The PTR-MS was operated with a drift tube voltage of 600 V and pressure of 1.9 mb and quantified isoprene emissions at m/z 69 as previously described [42].

### 4.6. Leaf Gas Exchange Analysis under Constant Environmental Conditions from Detached Branches Pretreated with [^13^C_2_]acetate or [^13^C]formate Solutions

For leaf gas exchange studies on detached branches pretreated with 10 mM [^13^C_2_]acetate or [^13^C]formate, a portable leaf photosynthesis system was utilized (Li6800 with small light source 6800-02, Li-Cor Biosciences, Lincoln, NE, USA). The subsampling port was used to divert a small percentage of the sample flow exiting the leaf chamber to the PTR-MS and CRDS for online measurements of isoprene and ẟ^13^CO_2_, respectively. The leaf chamber was operated under the following constant environmental conditions: 400 µmol s^−1^ hydrocarbon free air flowing through the leaf chamber, 30 °C leaf temperature, 420 ppm reference CO_2_, and 15 mmol mol^−1^ reference H_2_O. For the first part of the experiment, the light was held at 1000 µmol m^−2^ s^−1^ photosynthetically active radiation, before cycling off (30 min) and on (30 min) three times. Following an internal match of the reference and leaf sample infrared gas analyzers, leaf gas exchange observations of net photosynthesis (A, µmol m^−2^ s^−1^), transpiration (E, mmol m^−2^ s^−1^), and stomatal conductance (g_s_, mmol m^−2^ s^−1^) were recorded every 15 s.

For one DXSI and one leaf level experiment with 10 mM [^13^C_2_]acetate, real-time ẟ^13^CO_2_ observations using CRDS were compared with grab air samples and analyzed for ẟ^13^CO_2_ by isotope ratio mass spectrometry (IRMS). Samples (~65 mL) were collected in a sterile airtight syringe and then injected into 58 mL evacuated glass vials sealed with 20 mm-thick blue chlorobutyl rubber septa (Bellco Glass, Inc., Vineland, NJ, USA). Carbon isotope ratios of CO2 were measured in 5 mL aliquots extracted with an airtight syringe and injected into Helium-flushed 12 mL Labco exetainer vials. The CO_2_ was analyzed using a headspace autosampler (Gilson, Villiers-le-Bel, France) linked to a trace gas pre-concentrator interfaced to a Micromass JA Series Isoprime isotope ratio mass spectrometer (Micromass, Manchester, UK). The samples were transferred to a liquid nitrogen trap by flushing the Labco exetainer vial with ultra-pure He. By heating the liquid nitrogen traps, the CO_2_ was transferred and separated chromatographically from N_2_O on a Poraplot Q fused silica capillary column (30 m × 0.32 mm), and the carbon isotope ratios were measured in the mass spectrometer. Repeated injections of a laboratory standard associated with sample analysis yield δ^13^CO_2_ values of −33.79 ± 0.75‰ (1σ; *n* = 11).

## 5. Conclusions

Acetate fermentation has recently been demonstrated as a key mechanism of drought survival through the signaling properties of acetate including protein acetylation, with the amount of acetate produced directly correlating to survival. While the defense signaling properties of acetate have been the focus of previous studies via protein acetylation, here we demonstrate the potential for acetate in the transpiration stream of trees to be utilized as a respiratory substrate. The dynamic xylem solution injection method (DXSI) presented here allows for continuous injections of isotopically labeled metabolite solutions with programmable injection rates such that diurnal patterns of metabolic activity can be studied including differences in carbon allocation patterns during the day and night. The DXSI method has features of both the infusion method (low flow rates delivered over longer time periods) and the injection method (direct injection of liquid solutions into the xylem) and is therefore ideal for quantitative studies of leaf day respiration using isotopically labeled substances. While the majority of studies investigating day respiration in plants using isotopic labeling methods have employed common respiratory substrates during short duration labeling experiments with detached leaves (e.g., [1^3^C_1-6_]glucose and [^13^C_1-3_]pyruvate), we show that the alternative C1 and C2 respiratory substrates [^13^C_2_]acetate and [^13^C]formate are oxidized to ^13^CO_2_ within intact poplar trees with distinct differences in their rates of oxidation to ^13^CO_2_ and light influence. [^13^C]formate is rapidly converted to ^13^CO_2_ during transport in the transpiration stream, with ^13^CO_2_ efflux from canopy leaves increasing with transpiration during the day. In contrast, [^13^C_2_]acetate is efficiently transported to canopy leaves with a strong enhancement in leaf ^13^CO_2_ efflux at night when transpiration is reduced. Leaf gas-exchange experiments on detached branches pre-treated with [^13^C_2_]acetate in the light confirm the strong and reversible apparent light-inhibition of [^13^C_2_]acetate respiration. While acetate may not be able to directly enter the TCA cycle due to the potential lack of a mitochondrial acetyl-CoA synthetase, the results are consistent with the well characterized peroxisomal acetyl-CoA synthetase effectively integrating acetate into the glyoxylate cycle which is dependent on TCA cycle activity. The strong apparent suppression of [^13^C_2_]acetate respiration in the light cannot be explained by the well-known inhibition of mitochondrial PDH, and instead may be related to the light-inhibition of mitochondrial or peroxisomal CS. The results highlight the potential for an effective integration of acetate into the glyoxylate and TCA cycles and the light-inhibition of CS as an important regulatory mechanism in controlling the diurnal allocation of acetate between anabolic and catabolic processes.

## Figures and Tables

**Figure 1 plants-11-02080-f001:**
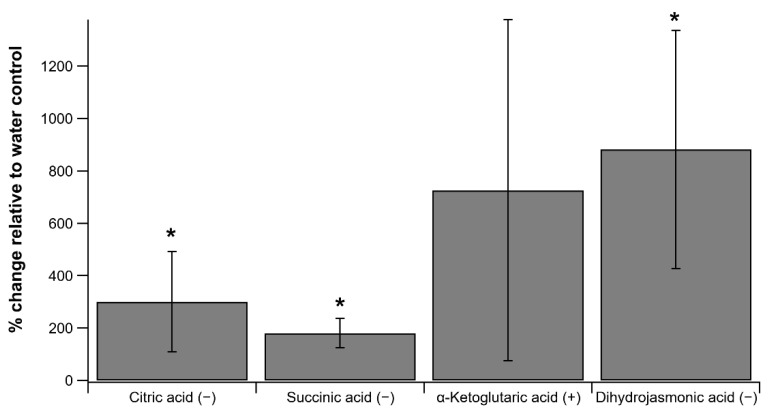
Average changes in leaf concentrations of TCA cycle intermediates Citric acid, Succinic acid, and α-Ketoglutaric acid relative to water-fed controls from three detached poplar branches treated with 10 mM acetate via the transpiration stream. Also shown are the relative changes in leaf concentrations of the phytohormone Dihydrojasmonic acid. Relative changes in leaf metabolite concentrations were determined by LC-MS (Liquid Chromatography Mass Spectrometry) operating in negative (−) and positive (+) modes. Vertical bars represent +/− 1 standard deviation and * indicates a statistically significant increase in the leaf concentration of TCA cycle intermediates relative to the control leaves (*p* < 0.05).

**Figure 2 plants-11-02080-f002:**
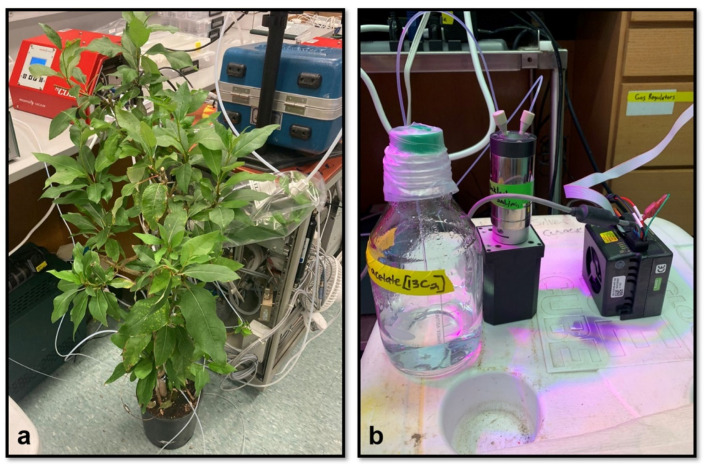
Images showing the (**a**) DXSI experimental set-up in the laboratory and (**b**) M6 pump and 500 mL solution reservoir.

**Figure 3 plants-11-02080-f003:**
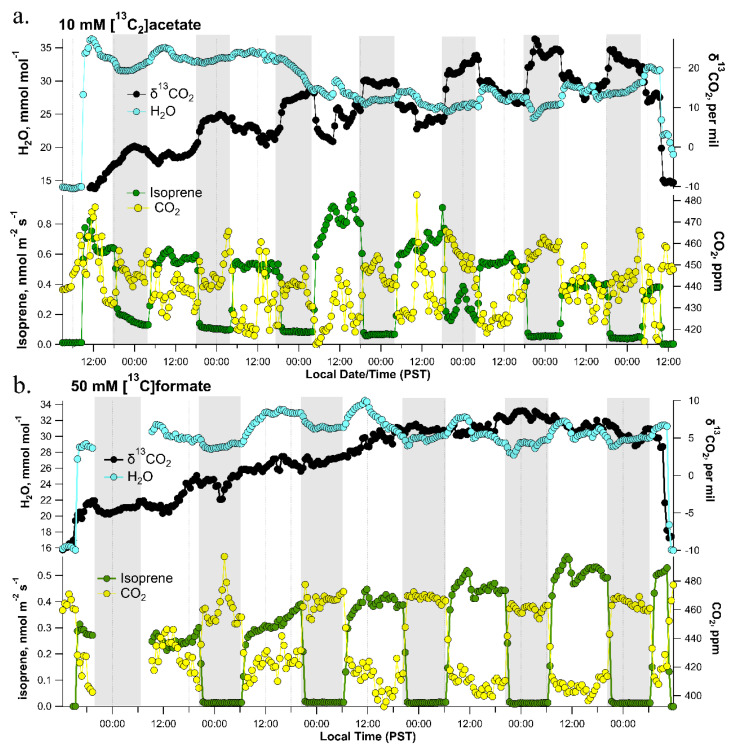
A 7-day time series of headspace CO_2_ and H_2_O water vapor concentrations, isoprene emissions, and headspace with δ^13^CO_2_ in a dynamic poplar branch enclosure during a DXSI experiment using (**a**) a 10.0 mM solution of [^13^C_2_]acetate and (**b**) a 50 mM solution of [^13^C]formate. The first and last part of the time series is an empty branch chamber.

**Figure 4 plants-11-02080-f004:**
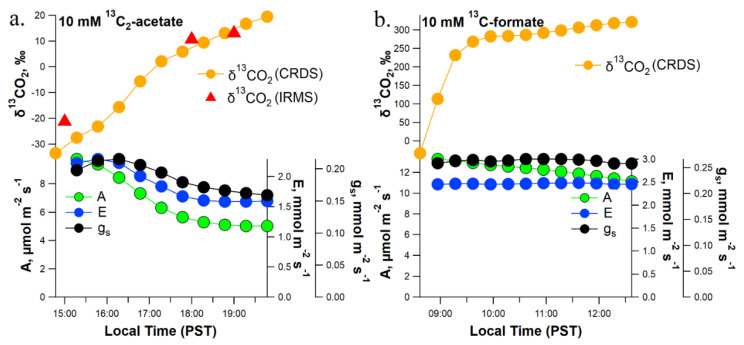
Example observations of leaf headspace ẟ^13^CO_2_ and leaf gas exchange parameters (A, E, g_s_) under constant environmental conditions in the light from detached poplar branches following 1-day delivery of a 10.0 mM solution of (**a**) [^13^C_2_]acetate and (**b**) [^13^C]formate via the transpiration stream. Note the stronger ^13^C-enrichment of headspace CO_2_ under [^13^C]formate relative to [^13^C_2_]acetate.

**Figure 5 plants-11-02080-f005:**
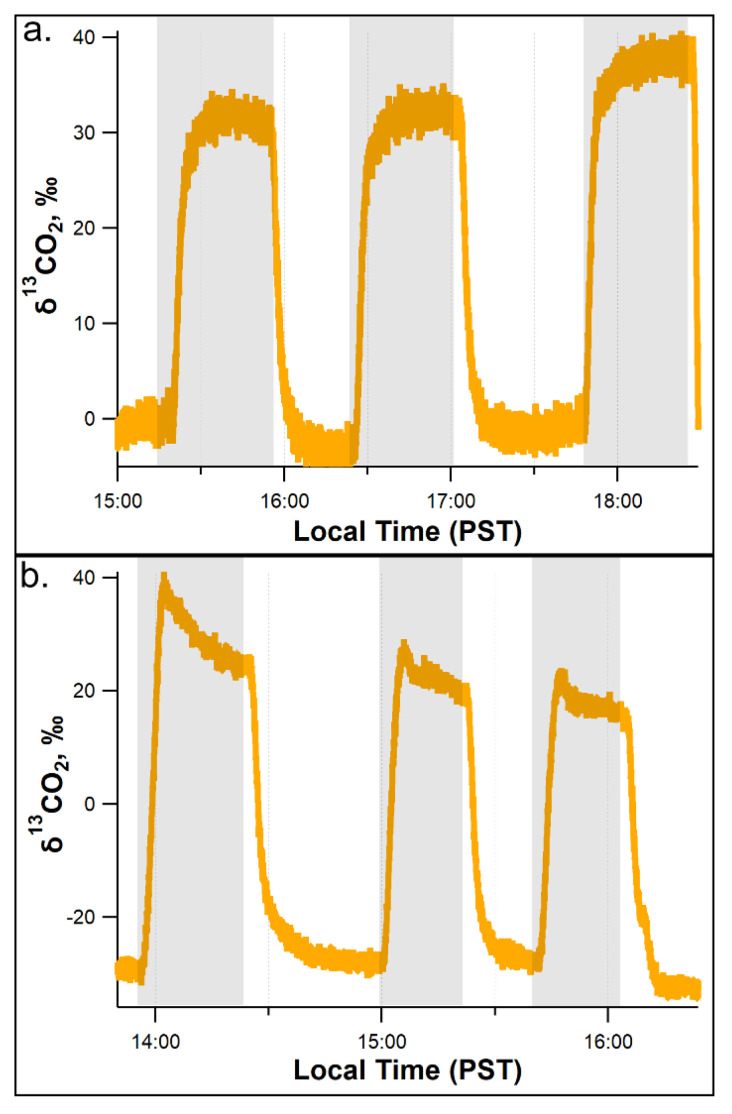
Rapid and reversible light inhibition of leaf headspace ẟ^13^CO_2_ from detached poplar branches following pretreatment with 10.0 mM [^13^C_2_]acetate via the transpiration stream. Two example time series (**a**,**b**) of leaf headspace ẟ^13^CO_2_ during three replicate light–dark and dark–light cycles under controlled environmental conditions. Note the reversible ^13^C-enrichment of headspace ẟ^13^CO_2_ upon leaf darkening.

**Figure 6 plants-11-02080-f006:**
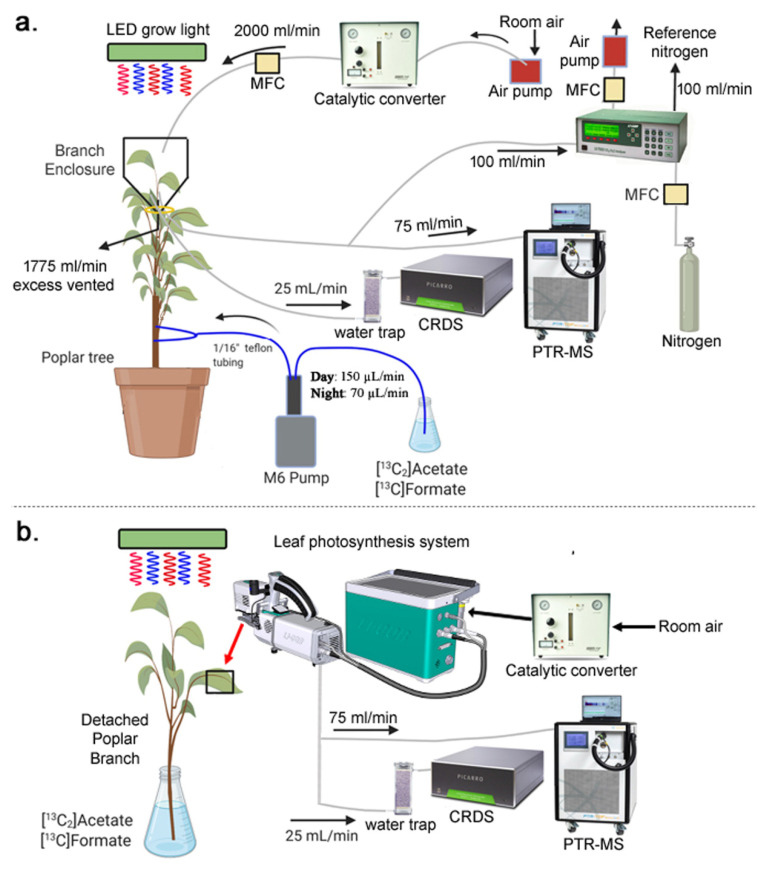
Graphical representation of whole tree/branch DXSI experiments and detached branch/leaf experiments with [^13^C_2_]acetate and [^13^C]formate solutions. The DXSI experimental setup (**a**) consists of an M6 pump programmed to deliver a diurnal solution dispensing method into the xylem of 70 µL/min (night) and 150 µL/min (day), and real-time branch gas exchange measurements of CO_2_, H_2_O, and isoprene fluxes together with ẟ^13^CO_2_ analysis under constant daytime lighting. Hydrocarbon-free room air was continuously delivered to a branch enclosure (2000 mL/min) to create a dynamic flow through gas-exchange system containing one of the top branches in the upper canopy (~2 m height) with 5–10 leaves. (**b**) Detached branches pretreated for 1 day with a 10 mM solution of [^13^C_2_]acetate or [^13^C]formate were studied for leaf gas exchange under controlled environmental conditions using a Li6800 system. Real-time branch and leaf isoprene emissions and ẟ^13^CO_2_ were quantified using PTR-MS and CRDS. CO_2_ and H_2_O fluxes were quantified using a Li6800 (leaf) or Li7000 (branch).

## Data Availability

All raw data generated during this study (PTR-MS, CRDS, IRMS, Li7000, Li6800, and LC-MS) can be acquired by contacting the corresponding author.

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
