# Peer review of "Light-Dependence of Formate (C1) and Acetate (C2) Transport and Oxidation in Poplar Trees"

_plants, 2022, doi:10.3390/plants11162080_

Round 1
Reviewer 1 Report
The manuscript is very interesting and very well conducted. The results are very valuable to understanding in greater depth the respiratory metabolism in poplar trees. The document is very well written and presented, and with some small improvements, it can be published in this important journal.

Author Response
Reviewer #1
Comment 1: The manuscript is very interesting and very well conducted. The results are very valuable to understand in greater depth the respiratory metabolism in poplar trees. The document is very well written and presented, with some small improvements it can be published in this important journal. I have the following suggestions to improve the manuscript
Response 1: We greatly thank reviewer 1 for his/her time to evaluate our article and the very supportive comments on the high quality of the writing, visual presentations, and science results/impact. In this paper, we introduce a new method to the plant sciences community (DXSI) while using it to study transport and light-dependent oxidation of alternative respiratory substrates formate and acetate in the transpiration stream of poplar trees. We have been extremely enthusiastic about the potential for the DXSI method to have a large impact in plant sciences as a new method to continuously deliver controlled flow rates of solutions of tracers, metabolites, pesticides, and many other substances for both research and commercial applications and have tried to balance the paper to describe this method and the new exciting information it revealed during our particular experiments on alternative respiratory metabolism of trees.
Comment 2: Abstract
Line 14: Populus trichocarpa on italic
The objective must coincide with the title, format is not included
Key words: change the keywords found in the title. I suggest “alternate respiratory pathways”
Response 2: These changes have been included in the new version of the manuscript. Note Populus trichocarpa was italicized in each instance in the manuscript.
Keywords found in the title removed and replaced with “alternate respiratory pathways”
Comment 3: Introduction Line 42: … usable chemical energy (ATP)
Response 3: The sentence now reads, “Aerobic respiration is a biochemical process requiring oxygen that occurs in living plant cells and provides usable chemical energy (ATP), reducing power (NADH), and carbon skeletons needed in numerous physiological processes including growth and development, reproduction, tissue maintenance/defense during stress, and senescence [1].”
Comment 4: Line 60-61: this idea is very important from the physiology of stress, can we mention other types of stress?
Response 4: We now include other types of stress by adding an additional sentence and reference, “JA signaling integrates with other phytohormone signaling pathways including salicylic acid, helps to coordinate plant defense responses to a wide array of abiotic and biotic stresses such as herbivory, pathogen infection, wounding, and high temperatures, freezing, drought, and salt stress [9].”
Comment 5: Lines 79-83: reorganize or complement the idea because the decrease in efficiency of PSII would occur when the amount of light that falls on the photosystems is greater than it can use. But inhibition of mitochondrial respiration by light can occur without a decrease in the efficiency of PSII.
Response 5:
Comment 6: Line 92: include reference
Response 6: Reference 3 now moved up to this location as suggested.
Comment 7: Line 144-146: summarize this paragraph, most of this information should go in the results chapter, the general objective should be clearer.
Response 7: This paragraph has been rewritten to improve the description of the general objectives of the new DXSI method development and tree respiration research described in the manuscript.
“In this study, we first hypothesize that two alternative respiratory pathways, including the oxidative C1 pathway and acetate fermentation (C2) are active in leaves and can be supported by formate (C1) and acetate (C2) derived from other tissues (e.g. roots and stems) and transported to leaves via the transpiration stream. Secondly, we hypothesize that due to the light inhibition of key respiratory enzymes like CS, light impacts the allocation of leaf acetate (C2) between anabolic (e.g. lipid biosynthesis) and catabolic (e.g. mitochondrial respiration) metabolism. We test these hypotheses in potted Populus trichocarpa trees by developing and applying the dynamic xylem solution injection (DXSI) method for continuous diurnal injections of [13C]formate and [13C2]acetate solutions into the xylem of poplar trees with controlled liquid injection flow rates together with continuous branch gas exchange measurements in the canopy of transpiration, net photosynthesis, isoprene emissions, and δ13CO2. These whole tree results are compared with leaf gas exchange observations on detached poplar branches pretreated with 10 mM [13C]formate and [13C2]acetate solutions and subjected to repeated light-dark cycles under controlled environmental conditions. The results are discussed in terms of the different timescales of formate and acetate metabolism in the transpiration stream of poplar trees and the apparent impact of light on regulating acetate allocation in leaves towards respiratory versus anabolic metabolism.”
Comment 8: The other aspects of the introduction seem very well addressed and sufficiently justified
Response 8: Thank you very much
Comment 9: Line 497: scientific name in cursive
Response 9: Scientific name now italicized throughout the manuscript
Comment 10: I suggest that the item 4.2 Relative quantification of leaf TCA cycle and jasmonate in acetate treated leaves, be placed after 4.4 Detached branch labeling
Response 10: Thank you for this suggestion. We now move the item 4.2 Relative quantification of leaf TCA cycle and jasmonate in acetate treated leaves, after 4.4 Detached branch labeling
Comment 11: Line 504: m-2 s-1 en superíndice
Response 11: Corrected, thank you
Comment 12: Line 520: change by ºC. min-1 in superíndice. H2O, 2 in subscript. Review these aspects throughout the document.
Response 12: These aspects have been corrected throughout the document.
Comment 13: Line 540-541: for more detail I suggest including a figure with a daily PAR curve
Response 13: We now clarify the PAR was held constant throughout the whole light period during the DXSI experiments: “For each of the four trees studied, they were first transported before sunrise from the UC Berkeley Oxford greenhouse to the nearby analytical laboratory (Lawrence Berkeley National Laboratory) and placed under an LED grow light (90 W Baisheng Semiconductor Lighting Co., Ltd.) supplying a constant 400-1000 µmol m-2 s-1 photosynthetically active radiation intensity at the top of the tree canopy (depending on position of canopy branch) during the daytime (6:00-18:00).”
Comment 14: 4.5 Gas exchange analysis: explain on which leaves the measurement was made and at what time of day. Indicate which parameters were measured (A, E gs)
Response 14: Details of gas exchange analysis for the two separate experiments (whole tree labeling with branch gas exchange and detached branch labeling with leaf gas exchange) are provided in this section 4.5.
Comment 15: Include statistical analysis
Response 15: We now include statistical analysis on figure 1 in the caption,
“…..*indicates a statistically significant increase in the leaf concentration of TCA cycle intermediates relative to the control leaves (p < 0.05). Note, the increase in the mean concentrations of α-Ketoglutaric acid were not statistically significant due to the relatively the high variation observed between the samples (N = 3).”
Comment 16: Line 150: scientific name in cursive. Include this subtitle: leaf concentrations of TCA cycle intermediates and jasmonates.
Response 16: Subtitle now included: “Leaf concentrations of TCA cycle intermediates and jasmonates”
Comment 17: Figure 1: is vertical barras in each average the standard deviation or standard error? Include in the title of the figure. Citric acid, ketoglutaric acid and dihydrojasmonic acid present very high variation, explain why within the discussion.
Response 17: Vertical bars in Figure 1 are now identified as +/1 one standard deviation in the caption.
Comment 18: Line 157-161: This information and figures 2 must go in materials and methods because it is a procedure. If they want to include it as a result, it must be indicated in materials and methods. Lines 293-296: this information is procedure must go in materials and methods.
Response 18: All increases in the mean leaf concentrations of TCA cycle intermediates were statistically significant relative to control leaves (p < 0.05), expect for α-Ketoglutaric acid due to the relatively the high variation observed between the samples (N = 3).
Comment 19: Supplementary material was not attached
Response 19: Supplementary material is now included
Comment 20: Dynamic Xylem Solution Injection (DXSI): I suggest to summarize this discussion because it is not the central topic of the paper, the other alternative is to include this aspect in the title of the paper.
Response 20: We struggled somewhat on where we could highlight the important DXSI method development as it is a major new technical contribution of the paper to the research community in addition to the science of alternative respiratory pathways being investigated with 13C-labeling. We now modify the title of the paper to reflect this to, "Dynamic Xylem Solution Injection (DXSI): A new Method to Investigate the Light-dependence of Formate (C1) and Acetate (C2) Transport and Oxidation in Trees”
Comment 21: Lines 384-405: as it stands, this paragraph is state of the art, they must return to the most important results and then discuss them
Response 21: This section has been improved as mentioned.
Comment 22: The discussion is well addressed, it correctly includes the explanations for the phytotoxic effect of the extracts on the growth inhibition of seedlings, and it compares with other similar studies.
Response 22: Thank you
Comment 23: Line 491-493: include bibliographic reference
Response 23: All references format carefully checked and included in bibliography
Comment 24: I suggest that the results and the discussion have the same subtitles
Response 24: We divided the discussion into two main sections. The first one discusses the new DXSI method and puts it into historical context, contrasting and comparing to previous solution injection methods. The second focuses on the light dependence of formate and acetate transport and metabolism in trees.
Reviewer 2 Report
The manuscript presents interesting finding particularly a useful insight into the mechanism of acetate allocation and its transport during day and night reaction in poplar trees. The labeling experiments were well prepared and reported and the results supports the conclusion. I will recommend the manuscript for publication.
Author Response
Reviewer #2
Comment 1: The manuscript presents interesting finding particularly a useful insight into the mechanism of acetate allocation and its transport during day and night reaction in poplar trees. The labeling experiments were well prepared and reported and the results supports the conclusion. I will recommend the manuscript for publication.
Response 1: Thank you very much to reviewer 2 for the support and time to review our article.
Reviewer 3 Report
The authors report an interesting analysis on the formation of formate and acetate in different physiological conditions of Populus trichocarpa. The authors' work is very precise and sophisticated. Using Populus trichocarpa leaves, they measure Jasmonate signaling and oxidative stress. They also demonstrate and confirm the role of acetate metabolism in regulating the response to stress by studying xylem transport. For this reason I suggest that the authors include this recent manuscript in the references of their paper:
Ashrafi, M. et al. Physiological and Molecular Aspects of Two Thymus Species Differently Sensitive to Drought Stress. BioTech 2022, 11, 8. https://doi.org/10.3390/biotech11020008
Furthermore, it would be important to specify better why the authors focused on the black poplar. Could these results also have a value for all crops of agronomic value?
Author Response
Reviewer #3
Comment 1: The authors report an interesting analysis on the formation of formate and acetate in different physiological conditions of Populus trichocarpa. The authors' work is very precise and sophisticated. Using Populus trichocarpa leaves, they measure Jasmonate signaling and oxidative stress. They also demonstrate and confirm the role of acetate metabolism in regulating the response to stress by studying xylem transport. For this reason I suggest that the authors include this recent manuscript in the references of their paper:
Ashrafi, M. et al. Physiological and Molecular Aspects of Two Thymus Species Differently Sensitive to Drought Stress. BioTech 2022, 11, 8. https://doi.org/10.3390/biotech11020008
Response 1: We greatly thank reviewer 3 for the supportive and helpful comments which we have addressed in the new version of the manuscript. We now cite the very relevant paper indicated by the reviewer, which helps establish the role of acetic acid in central metabolism and its importance during the plant drought response.
“Recently, plant tissue concentrations of acetate were shown to dramatically increase due to the activation of acetate fermentation during drought stress [7, 8] and water stress was confirmed to to enhance the gene expression of specific enzymes involved in acetate fermentation (e.g. pyruvate decarboxylase) and acetate activation to acetyl-CoA (acetyl-CoA synthethetase) [9].”
Comment 2: Furthermore, it would be important to specify better why the authors focused on the black poplar. Could these results also have a value for all crops of agronomic value?
Response 2: We now include a statement in the Materials and Methods section under 4.1 plant material about why we focused on back poplar. The question about the DXSI method and results having importance for all crops of agronomic value is extremely interesting, but deserves a separate study as it is outside of the scope of the present work.
“We focused on the fast-growing genera Populus as it is widely distributed across many temperate and cold regions across the word and is actively being investigated for afforestation efforts and as renewable sources of biofuels and bioproducts. Moreover, with the completion of genome sequencing, Populus trichocarpa is considered highly amendable to future genetic engineering studies aimed at altering carbon allocation patterns of woody plants”.